# Molecular mechanism underlying regulation of *Arabidopsis* CLCa transporter by nucleotides and phospholipids

Zhao Yang [1,2,6], Xue Zhang [1,6], Shiwei Ye[2,3,6], Jingtao Zheng[4], Xiaowei Huang [1,2], Fang Yu[4], Zhenguo Chen [5] ✉, Shiqing Cai [3] ✉ & Peng Zhang [1] ✉

Chloride channels (CLCs) transport anion across membrane to regulate ion homeostasis and acidification of intracellular organelles, and are divided into anion channels and anion/proton antiporters. *Arabidopsis thaliana* CLCa (AtCLCa) transporter localizes to the tonoplast which imports $NO_3^-$ and to a less extent $Cl^-$ from cytoplasm. The activity of AtCLCa and many other CLCs is regulated by nucleotides and phospholipids, however, the molecular mechanism remains unclear. Here we determine the cryo-EM structures of AtCLCa bound with $NO_3^-$ and $Cl^-$, respectively. Both structures are captured in ATP and PI(4,5)P$_2$ bound conformation. Structural and electrophysiological analyses reveal a previously unidentified N-terminal β-hairpin that is stabilized by ATP binding to block the anion transport pathway, thereby inhibiting the AtCLCa activity. While AMP loses the inhibition capacity due to lack of the β/γ-phosphates required for β-hairpin stabilization. This well explains how AtCLCa senses the ATP/AMP status to regulate the physiological nitrogen-carbon balance. Our data further show that PI(4,5)P$_2$ or PI(3,5)P$_2$ binds to the AtCLCa dimer interface and occupies the proton-exit pathway, which may help to understand the inhibition of AtCLCa by phospholipids to facilitate guard cell vacuole acidification and stomatal closure. In a word, our work suggests the regulatory mechanism of AtCLCa by nucleotides and phospholipids under certain physiological scenarios and provides new insights for future study of CLCs.

CLC proteins, composed of anion channels and anion/proton antiporters, transport $Cl^-$, $NO_3^-$, or $F^-$ across membranes, and are ubiquitous in all kingdoms[1]. In mammals, two different CLCs have been defined, $Cl^-$ channels (CLC-1/2/Ka/Kb) located at the plasma membrane and $2Cl^-/1H^+$ antiporters (CLC-3/4/5/6/7) located at the internal membrane, which are involved in the regulation of muscle excitability, transepithelial $Cl^-$ transport, and acidification of intracellular organelles, and mutations of CLCs lead to serious diseases[2]. In plants, CLCs

[1]National Key Laboratory of Plant Molecular Genetics, Center for Excellence in Molecular Plant Sciences, Institute of Plant Physiology and Ecology, Chinese Academy of Sciences, Shanghai 200032, China. [2]University of Chinese Academy of Sciences, Beijing 100039, China. [3]Center for Excellence in Brain Sciences and Intelligence Technology, Institute of Neuronscience, Chinese Academy of Sciences, Shanghai 200031, China. [4]Shanghai Key Laboratory of Plant Molecular Sciences, College of Life Sciences, Shanghai Normal University, Shanghai 200234, China. [5]The Fifth People's Hospital of Shanghai, Institutes of Biomedical Sciences, School of Basic Medical Sciences, Fudan University, Shanghai 200032, China. [6]These authors contributed equally: Zhao Yang, Xue Zhang, Shiwei Ye. ✉e-mail: zhenguochen@fudan.edu.cn; sqcai@ion.ac.cn; pengzhang01@cemps.ac.cn

are known to perform several functions including nutrient storage, stomatal regulation, photosynthesis, embryonic development, and abiotic and biotic tolerance[3–14]. Seven CLCs have been identified in *Arabidopsis thaliana* which are all located in the endomembrane system: CLCa−c and CLCg in tonoplast[4–9], CLCd and CLCf in Golgi vesicles[10–13], and CLCe in thylakoid membrane of chloroplasts[13,14]. AtCLCa is highly expressed in mesophyll cells where it functions as a $2NO_3^-/1H^+$ antiporter transporting $NO_3^-$ into the vacuole for storage[3,4]. Meanwhile, AtCLCa is abundant in guard cells where it is not only involved in stomatal opening in response to light, but also required for stomatal closure induced by ABA[5,6].

The transport mechanism of CLCs has been well established by structural studies which revealed a homodimeric architecture with each protomer bearing a separate ion transport pathway and three consecutive substrate-binding sites referred to as the external site ($S_{ext}$), central site ($S_{cen}$) and internal site ($S_{int}$)[15–25]. A highly conserved glutamate residue near $S_{ext}$, called the gating glutamate ($Glu_{gate}$) or external glutamate ($Glu_{ex}$) (Supplementary Fig. 1), serves as a gate for the $Cl^-$ transport in both CLC channels and transporters[16,17], as well as for $H^+$ transit in CLC transporters[6,26–29]. In the cytosolic side, away from the $Cl^-$ transport pathway, there is a proton glutamate or internal glutamate ($Glu_{in}$) coupled with $H^+$ transport, which is highly conserved among the CLC transporters[29–32] (Supplementary Fig. 1). The anion selectivity is determined by a single residue near $S_{cen}$, a serine in most $Cl^-$ selective CLCs, however, the serine is replaced by a proline in $NO_3^-$ selective AtCLCa (Supplementary Fig. 1), and the conversion between serine and proline can alter the anion selectivity of CLCs[29,33,34].

A number of studies reported that the activity of eukaryotic CLCs were regulated by nucleotides and phospholipids. The activity of CLCs was regulated by nucleotides through binding to the C-terminal cystathionine-β-synthetase (CBS) domain[35–41]. Mammalian CLC-1/2 channels are inhibited by ATP[36–38], while the activity of CLC-3/4/5 transporters is enhanced by ATP[39,40]. As for AtCLCa, ATP could significantly reduce the transport activity while AMP could compete with ATP to compromise the inhibition which adjusted the $NO_3^-$ flow into the vacuole depending on photosynthetic efficiency[41]. The ATP-

binding site in CBS domain is unambiguous[23,25,35], however, the molecular mechanism of how nucleotides regulate the activity of CLCs remains elusive. In addition, the activity of CLCs was regulated by phosphatidylinositol-3,5-bisphosphate ($PI(3,5)P_2$), a low-abundance signaling lipid associated with vacuolar and lysosomal membranes in eukaryotic cells[42,43]. In *Arabidopsis*, the AtCLCa activity could be inhibited by $PI(3,5)P_2$, which contributed to vacuolar acidification, a process required for ABA-induced stomatal closure in guard cells[42,44]. In human, the CLC-7 activity could also be attenuated by $PI(3,5)P_2$, which adjusted lysosomal pH[43,45]. However, the underlying regulatory mechanism is also mysterious.

Here we determine the cryo-EM structures of AtCLCa transporter bound with $Cl^-$ and $NO_3^-$, respectively. Both structures are captured in ATP and $PIP_2$ bound conformation. The high-resolution structures demonstrate the binding sites of ATP and $PIP_2$, and their interaction details with AtCLCa. Structural-based analyses suggest how the AtCLCa activity was regulated by ATP and $PIP_2$.

## Results

### Overall structure

AtCLCa proteins were expressed, purified, and reconstituted into nanodiscs for structural analysis. To obtain the structure of AtCLCa binding with $Cl^-$ or $NO_3^-$, proteins were purified in buffer containing 150 mM NaCl or $NaNO_3$, respectively. The cryo-EM structure of AtCLCa in complex with $Cl^-$ was determined to 3.0 Å (AtCLCa-$Cl^-$), while that in complex with $NO_3^-$ was determined to 3.2 Å (AtCLCa-$NO_3^-$) (Supplementary Fig. 2 and Supplementary Table 1). The density maps of both structures were of sufficient quality for model building. Superposition of AtCLCa-$Cl^-$ and AtCLCa-$NO_3^-$ structures shows a root mean square deviation (RMSD) of 0.8 Å over 641 Cα atoms, and superposition of the recently published AtCLCa structure with our AtCLCa-$Cl^-$ (over 627 Cα atoms) or AtCLCa-$NO_3^-$ (over 701 Cα atoms) structures shows a RMSD of 0.9 Å and 0.6 Å, respectively.

AtCLCa forms a homodimer (Fig. 1a−c). Each protomer contains a transmembrane domain (TMD) formed by helices αB to αS and two cytoplasmic CBS domains (Fig. 1d, e). Both the TMD and CBS domains

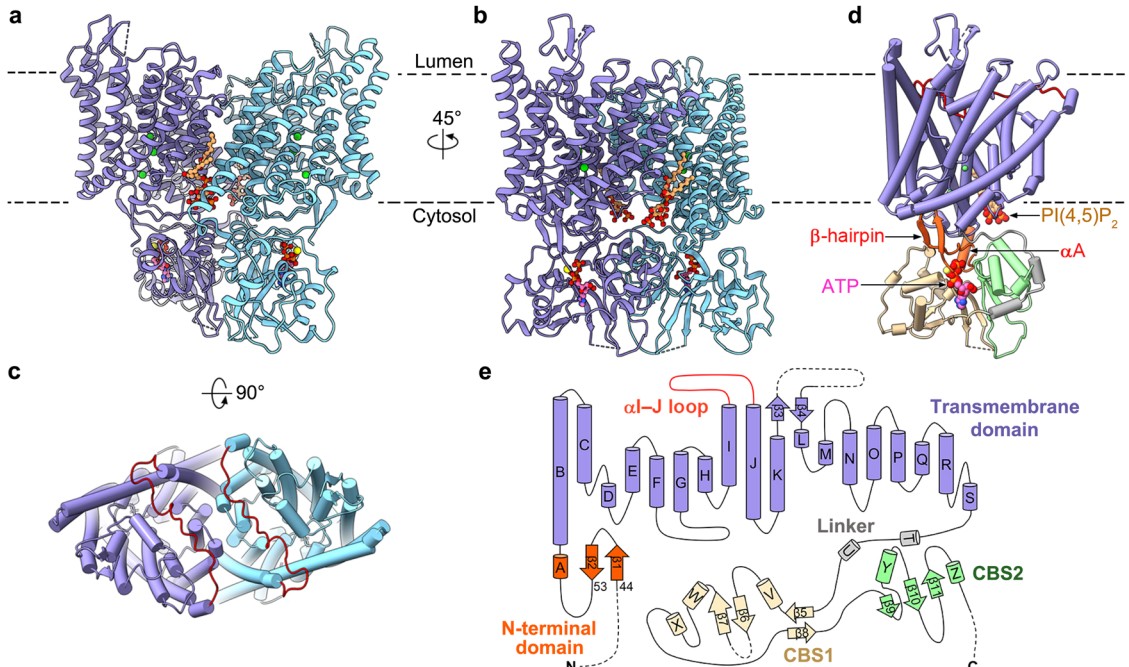

**Fig. 1 | Overall structure. a, b** Structure of the AtCLCa-$Cl^-$ dimer in side view (**a**) and right turn 45° view (**b**). One protomer is colored blue and the other cyan. $Cl^-$, green spheres; ATP, pink balls and sticks; $PI(4,5)P_2$, beige balls and sticks; $Mg^{2+}$, yellow spheres. **c** Top view of AtCLCa-$Cl^-$. The αI−J loop is colored red. **d** An AtCLCa protomer in the same orientation as in (**b**) and colored by domains. NTD, orange red; TMD, blue; αI−J loop, red; Linker, gray; CBS1, beige; CBS2, green; ATP, pink spheres; $PI(4,5)P_2$, beige spheres. **e** Topology of the AtCLCa protomer colored by domain as in (**d**).

contribute to the dimer interface (Fig. 1a, b), which were usually found in eukaryotic CLC structures[17,18,20,23–25]. Besides, an N-terminal αA helix and a preceding loop (residues 54–64) sandwiched between the TMD and the CBS domains are found in each protomer, which was also observed in CLC-7 and recently published AtCLCa structures[23–25] (Fig. 1d, Supplementary Fig. 3a–c). Moreover, a β-hairpin structure comprised of residues 44–53 is observed in front of the N-terminal loop in each AtCLCa-Cl⁻ protomer (Supplementary Fig. 2k), hence, the N-terminal domain (NTD) of AtCLCa is constituted by the β-hairpin, the αA helix and the connecting loop (Fig. 1d, e). However, the β-hairpin is either disordered in the AtCLCa-NO₃⁻ structure or not observed in reported CLC structures including the recently published AtCLCa[17,18,20,21,23–25] (Supplementary Fig. 3a–c). One molecule of ATP and one molecule of PI(4,5)P₂ are found binding with each protomer in both structures (Fig. 1d, Supplementary Fig. 2k, l). Since we did not add ATP or PI(4,5)P₂ during purification, they must be co-purified with AtCLCa from the yeast expression system.

Although the overall structure of AtCLCa-Cl⁻ and AtCLCa-NO₃⁻ is similar, significant local conformational differences are observed on the luminal side, represented by the αI–J loop (304–327) (Fig. 2a, b, Supplementary Fig. 3d). In the AtCLCa-Cl⁻ structure, αI–J loop covers the ion transport pathway like a lid from the luminal side (Fig. 2b, c), whereas in the AtCLCa-NO₃⁻ structure, αI–J loop moves away from the ion transport pathway and leaves the ion transport pathway open to the lumen (Fig. 2b, d). Comparing our two structures with the previously reported CLC-ec1 and HsCLC-1 structures, we found that the αI–J loop in the AtCLCa-NO₃⁻ structure but not the AtCLCa-Cl⁻ structure adopts a similar conformation as that in CLC-ec1 and HsCLC-1, and all leave the transport pathway in open conformation (Supplementary Fig. 3f). Therefore, we conclude that the AtCLCa-NO₃⁻ structure was captured in a luminal pathway open conformation (same conformation was captured in the recently published AtCLCa[25]) (Fig. 2d, Supplementary Fig. 3e, g), while the AtCLCa-Cl⁻ represented a luminal pathway close conformation (Fig. 2c).

## Transport pathway

The TMD of each AtCLCa protomer contains an independent ion transport pathway, which is bifurcated on the cytoplasmic side, one for anion transport and the other for H⁺ transport (Fig. 3a, b, d, e). The anion transport pathway is open on both the luminal and cytosolic sides in the AtCLCa-NO₃⁻ structure (Fig. 3d), but it is closed by the αI–J loop on the luminal side and only partially open on the cytosolic side due to the presence of the NTD β-hairpin in the AtCLCa-Cl⁻ structure (Fig. 3a). In the AtCLCa-Cl⁻ structure, three conserved Cl⁻-binding sites are well-preserved (Fig. 3b, Supplementary Fig. 4a). The Cl⁻ in S_ext is coordinated by the main-chain nitrogen of Gly201 on helix αF, as well as Leu479, Phe480, and Leu481 on helix αO. The Cl⁻ in S_cen is coordinated by the side chain of Tyr564 on helix αS and main-chain nitrogen of Leu479 on helix αO. The Cl⁻ in S_int is coordinated by the main-chain nitrogen of Gly161 and the side chain of Glu164 on helix αD (Fig. 3c). While in the AtCLCa-NO₃⁻ structure, two of the three substrate-binding sites, S_cen and S_int were found of fair densities, which likely corresponds to the bound NO₃⁻ (Fig. 3e, Supplementary Fig. 4b). The NO₃⁻ in S_cen is coordinated by hydrophobic residues, including Pro160 and Ile162 on helix αD, Leu479 on helix αO and Met522 on helix αP, and forms hydrogen bond with the side chain of Tyr564 on helix αS. The NO₃⁻ in S_int forms hydrogen bonds with the side chains of Glu164 on helix αD and Lys571 on helix αS (Fig. 3f).

The electrophysiological properties of AtCLCa have been well studied in plant vacuoles and *Xenopus* oocytes[4–6,29,34,41,44]. It has been extensively reported that Pro160 determines the NO₃⁻ preference of AtCLCa, which was replaced by a serine in Cl⁻ selective CLCs (Supplementary Figs. 1, 4d, e), and the conversion between serine and proline can alter the anion selectivity of CLCs[29,33,34]. Comparing structures of AtCLCa-NO₃⁻, AtCLCa-Cl⁻ and other reported CLCs, differences are observed in the binding sites of Cl⁻ and NO₃⁻ (Supplementary Fig. 4c, e). It is notable that Cl⁻ in S_cen forms hydrogen bond with the side chain of the conserved serine in Cl⁻ selective CLCs, but this hydrogen bond cannot be formed between anion and Pro160 in

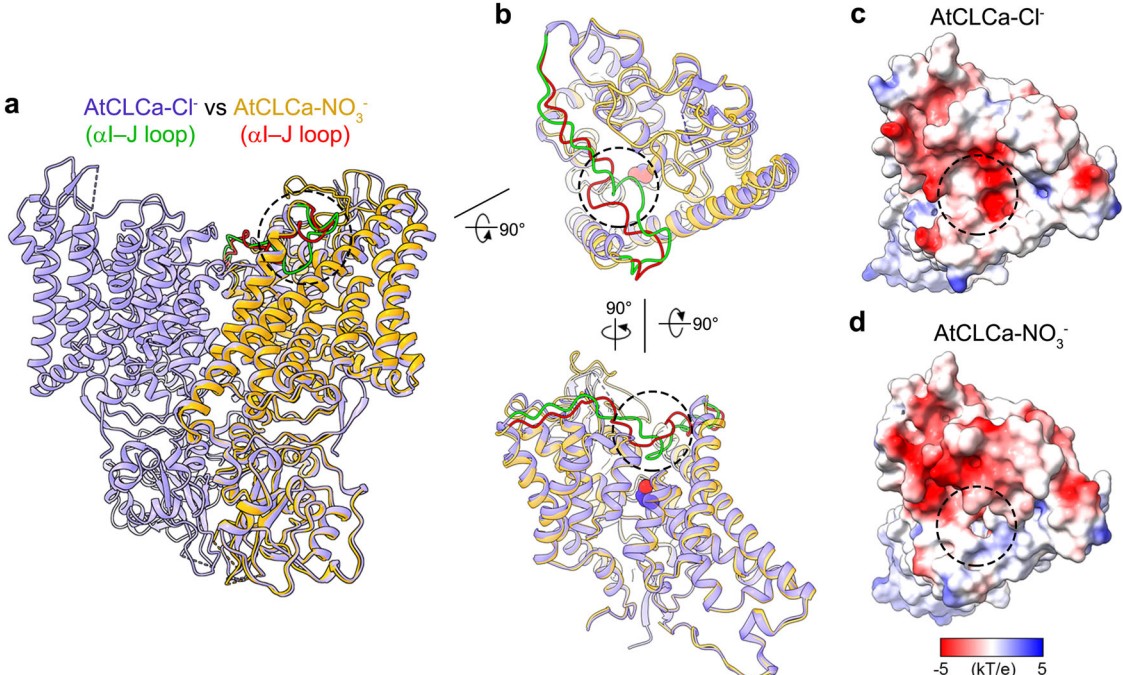

**Fig. 2 | Conformational difference on the luminal side. a** Comparison of AtCLCa-Cl⁻ (blue) and AtCLCa-NO₃⁻ (golden). The AtCLCa-NO₃⁻ protomer was superimposed onto one protomer of AtCLCa-Cl⁻ dimer. The αI–J loop is colored green in AtCLCa-Cl⁻ and red in AtCLCa-NO₃⁻. **b** Comparison of the AtCLCa-Cl⁻ and AtCLCa-NO₃⁻ protomer. The CBS domains are hidden for clearance. The dashed circle indicates segments of αI–J loop near the outlet of ion transport pathway in the luminal side. The side chain of gating glutamate Glu203 is shown as spheres. **c, d** Electrostatic potential surface of the AtCLCa-Cl⁻ (**c**) and AtCLCa-NO₃⁻ (**d**) protomer. The dashed circle indicates the outlet of ion transport pathway in luminal side.

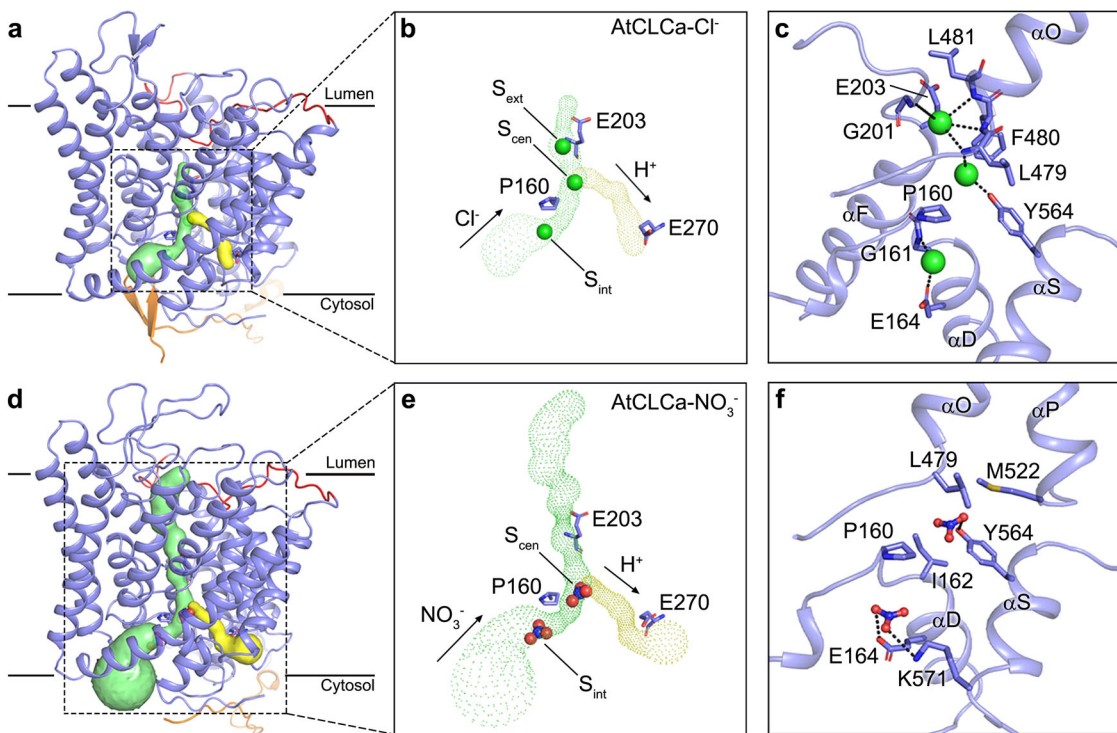

**Fig. 3 | Transport pathway. a, d** The ion transport pathway in AtCLCa-Cl⁻ (**a**) and AtCLCa-NO₃⁻ (**d**) protomer. Protomers are shown in side view with CBS domains hidden. NTD and TMD are colored orange and blue, respectively. The αI–J loop is colored red. The bifurcated ion transport pathways are shown as solid surface, green for Cl⁻ (**a**) or NO₃⁻ (**d**) transport pathway and yellow for H⁺ transport pathway, calculated by Caver[58]. **b, e** A zoom-in view of the bifurcated ion transport pathway of AtCLCa-Cl⁻ (**b**) and AtCLCa-NO₃⁻ (**e**) at the same magnification, displayed as meshes. Arrows indicate the direction of ion transport in the physiological state. Cl⁻ (**b**) and NO₃⁻ (**e**) are shown as spheres. Three critical residues near the pathway are shown as sticks. **c, f** Anion-binding sites. Cl⁻ (**c**) and NO₃⁻ (**f**) are shown as spheres and balls and sticks, respectively. Hydrogen bonds are shown as black dashed lines.

AtCLCa (Supplementary Fig. 4e). These may explain the selectivity of AtCLCa for NO₃⁻.

Two conserved glutamates play indispensable roles in H⁺-coupled anion transport in CLC transporters[6,26–32] (Supplementary Fig. 1). The gating glutamate controls the ion access and the coupling of external H⁺, while the proton glutamate is indispensable for the coupling of cytosolic H⁺. The gating glutamate E203A mutant of AtCLCa abolished H⁺ but not anion transport[6,29], while the proton glutamate E270A mutant of AtCLCa eliminated both H⁺ and anion transport[29]. The side chain of gating glutamate oscillates between different conformations[16,17,20,22,23,25] (Supplementary Fig. 4f). In our AtCLCa structures, the gating glutamate Glu203 adopts the "up" conformation whose side chain moves out of the binding sites and points into the vacuolar lumen, and the proton glutamate Glu270 is located at the cytoplasmic outlet of the H⁺ transport pathway near the dimer interface (Fig. 3b, e, Supplementary Fig. 4f).

### ATP-binding site and inhibition mechanism

The β-hairpin of NTD, firstly identified in our structure, is located below the TMD and occupies the cytoplasmic entrance of the anion transport pathway in each AtCLCa-Cl⁻ protomer (Fig. 4a, Supplementary Fig. 5a, b). Two small side-chain residues Gly48 and Ala49 constitute the turn of the β-hairpin, and form hydrogen-bonding interactions with the side chains of Glu164 from αD, Lys571 from helix αS and Lys370 from helix αK (Fig. 4b). The unique location of the β-hairpin of NTD and its interaction with TMD suggests that it may serve as a "plug" to block the anion transport pathway to regulate the AtCLCa activity.

Below the NTD, a non-protein density peak was observed between the NTD and the CBS domains in each protomer and was interpreted as Mg²⁺-bound ATP molecule (Fig. 4a–c, Supplementary Fig. 5a, 5c). The

ATP molecule forms extensive interactions with NTD and CBS domains (Fig. 4c), as observed in CLC-7 and recently published AtCLCa structures[23,25]. Specifically, two residues Glu55 and Ser56 protrude from the NTD loop interacts with ATP. Residue Glu55 indirectly interacts with three phosphate groups of ATP through forming ionic bonds with Mg²⁺, and the adjacent residue Ser56 forms hydrogen bonds with the γ-phosphate groups of ATP. A number of residues from CBS1 and CBS2 domains are involved in ATP coordination. Residues His620 and Ile748 stacks with the adenine base of ATP, which is also stabilized by Val600 and Ala622. Residues Lys596 and Asp753 form hydrogen bonds with the hydroxyl group of the ribose moiety of ATP. Residues His620, Asn621, Arg729, and His730 form hydrogen bonds with the phosphate groups of ATP (Fig. 4c). The structural information suggests that ATP may function as a molecular glue to stick the NTD and CBS domains together, thus stabilizing the conformation of the β-hairpin to block the anion transport pathway of AtCLCa. This may give a molecular explanation of the ATP inhibition toward AtCLCa[5,41].

To verify this possibility, we performed whole-cell patch-clamp experiments in HEK293T cells (Supplementary Fig. 6). As previously shown[4,29,34], expression of AtCLCa yielded significant currents well above background levels with a higher NO₃⁻ permeability than Cl⁻ (Supplementary Fig. 6a, d, e). In this configuration, the intracellular side of HEK293T cells is equivalent to the external side of plant vacuoles. Application of 5 mM ATP to the cytosolic solution led to a decrease of approximately half of the NO₃⁻ currents induced by wild-type AtCLCa (Fig. 4d). ATP-γ-S induced an inhibition comparable with ATP, indicating that the ATP inhibition is independent of ATP hydrolysis (Supplementary Fig. 6f). ADP produced a weak inhibition while AMP induced a negligible inhibition (Fig. 4d), suggesting that the β/γ-phosphate of adenine nucleotides are important for their inhibition on AtCLCa. When both AMP and ATP were added together, the currents

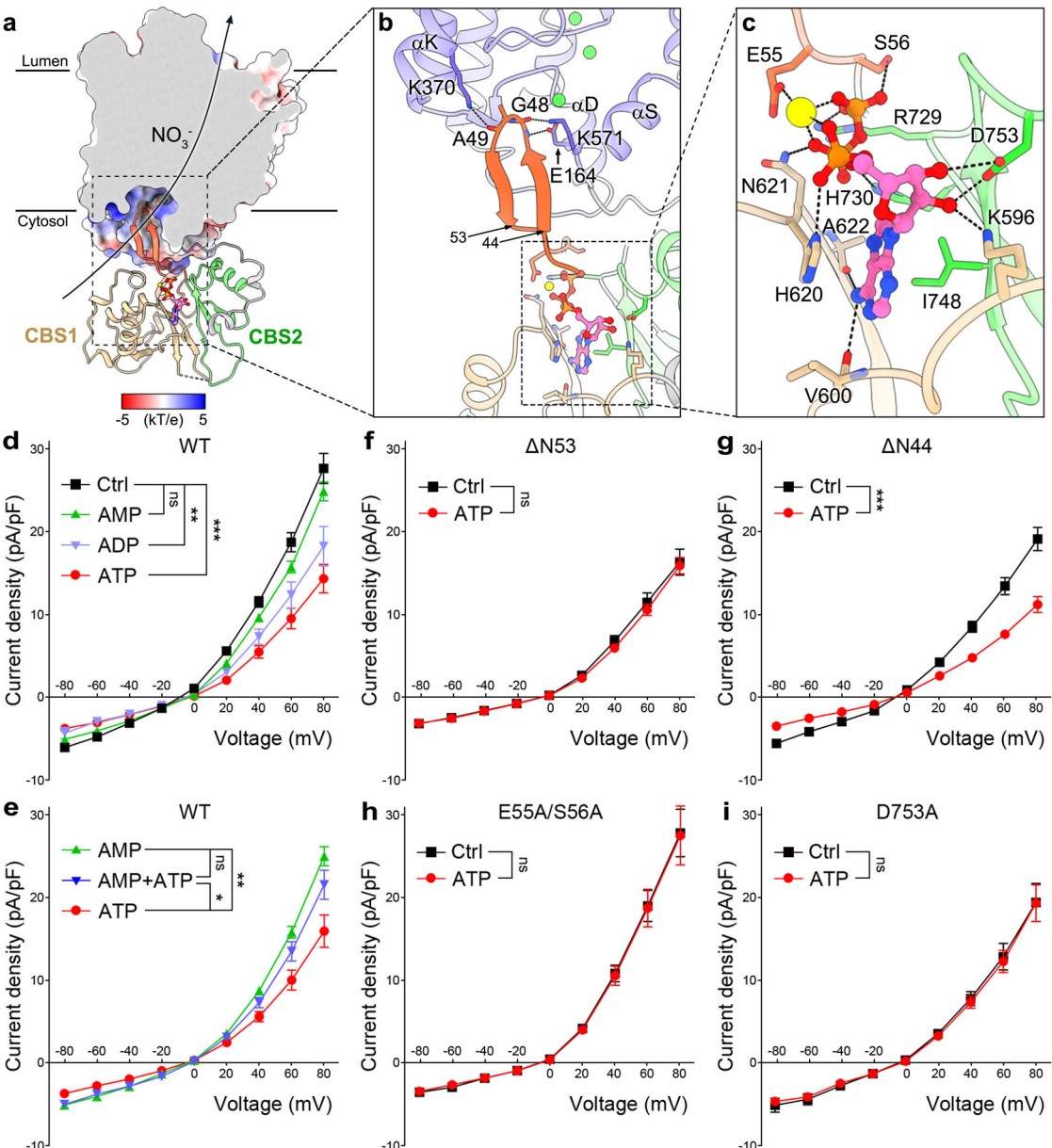

**Fig. 4 | ATP-binding site and inhibition mechanism. a** A cut-open view of AtCLCa-Cl⁻ protomer. TMD is shown as a cut-open view of the electrostatic potential surface. Curved arrow indicates the direction of $NO_3^-$ transport in the physiological state. NTD, orange; linker, gray; CBS1, beige; CBS2, green; ATP, pink balls and sticks; Cl⁻, green sphere; $Mg^{2+}$, yellow sphere. The dashed box is represented to show the N-terminal β-hairpin and ATP-binding site. **b** A zoom-in view of the interaction between the β-hairpin and TMD. Hydrogen bonds are shown as black dashed lines. The dashed box is represented to show the ATP-binding site. **c** A zoom-in view of ATP-binding site. **d** I–V curves of wild-type AtCLCa with different adenine nucleotides at a concentration of 5 mM added to the pipette solution. $n = 20$ (Ctrl, nucleotide free), n = 22 (ATP), $n = 19$ (ADP), $n = 16$ (AMP). $P < 0.0001$ (ATP vs Ctrl), $P = 0.008$ (ADP vs Ctrl), $P = 0.1391$ (AMP vs Ctrl). **e** I–V curves of wild-type AtCLCa with 5 mM AMP and 5 mM ATP added separately or together to the pipette solution.

$n = 20$ (ATP), $n = 16$ (AMP), $n = 16$ (ATP + AMP). $P = 0.0011$ (ATP vs AMP), $P = 0.011$ (ATP vs ATP and AMP), $P = 0.5689$ (AMP vs ATP and AMP). **f–i** I–V curves of AtCLCa ΔN53 mutant (**f**), ΔN44 mutant (**g**), E55A/S56A mutant (**h**), and D753A mutant (**i**) with 5 mM ATP added to the pipette solution. For ΔN53 mutant (**f**), $n = 26$ (Ctrl), $n = 26$ (ATP), $P = 0.8163$. For ΔN44 mutant (**g**), $n = 23$ (Ctrl), $n = 20$ (ATP), $P < 0.0001$. For E55A/S56A mutant (**h**), $n = 25$ (Ctrl), $n = 26$ (ATP), $P = 0.7026$. For D753A mutant (**i**), $n = 23$ (Ctrl), $n = 24$ (ATP), $P = 0.7239$. For (**d–i**), the whole-cell currents were evoked by clamping the HEK293T cells for 2-s voltage pulses from −80 to +80 mV in 20-mV steps followed by a repolarizing step to −80 mV, and the nitrate bath solution was used. Data are mean ± s.e.m. Two-tailed $t$-tests were used for significance analysis of the current density at +80 mV. *$P < 0.05$, **$P < 0.01$, ***$P < 0.001$, ns means no significance difference. Source data are provided as a Source data file.

decrease was smaller compared to ATP alone (Fig. 4e), suggesting that AMP could compete with ATP to prevent its inhibition toward AtCLCa. These data are consistent with previous studies of AtCLCa in plant vacuoles[5,41], which also suggests that HEK293T system can be used for AtCLCa study.

To test the function of the β-hairpin for the ATP inhibition, the β-hairpin of AtCLCa was deleted in the ΔN53 mutant, which was transfected to HEK293T cells. Comparing to the wild type, the $NO_3^-$ currents

in ΔN53 mutant are not influenced by the addition of ATP, although a slightly decrease of current densities is observed in the absence of ATP (Fig. 4f). In contrast, the ΔN44 mutant, which contains the N-terminal 44-residue deletion but retains the β-hairpin, exhibits similar inhibition pattern to wild-type AtCLCa upon ATP addition (Fig. 4g). These results further support the structural data, and both suggest that the NTD β-hairpin is required to retain the ATP inhibition toward AtCLCa. We further examined the effect of mutations in the ATP-binding site.

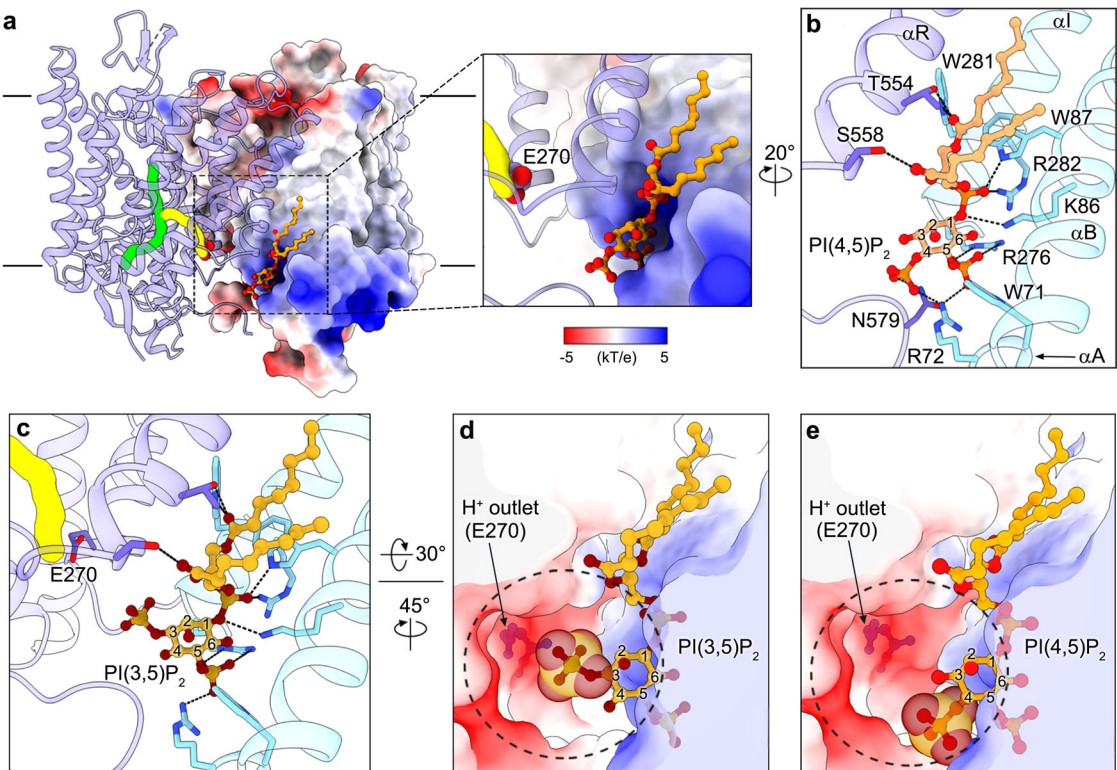

**Fig. 5 | PIP₂-binding site and possible inhibition mechanism. a** PI(4,5)P₂ binds to the dimer interface of AtCLCa. AtCLCa-Cl⁻ dimer is shown in side view with CBS domains hidden. Protomer A and B are shown as electrostatic surface model and ribbon structure, respectively. The anion and H⁺ transport pathways are shown as green and yellow solid surfaces, respectively. Inset: a zoom-in view of the PI(4,5)P₂-binding site. PI(4,5)P₂ is shown as balls and sticks. The side chain of proton glutamate Glu270 is shown as spheres. **b** The interaction between PI(4,5)P₂ and AtCLCa-Cl⁻. Protomer A and B are colored in cyan and blue, respectively. Hydrogen bonds are shown as black dashed lines. **c** Modeled coordination of PI(3,5)P₂ into the PI(4,5)P₂-binding site of AtCLCa. **d** The C3-phosphate of PI(3,5)P₂ occupies the H⁺ outlet. AtCLCa-Cl⁻ is shown as a cut-open view of the electrostatic potential surface. The dashed circle indicates the H⁺ outlet. The C3-phosphate of PI(3,5)P₂ is shown as spheres. **e** The C4-phosphate of PI(4,5)P₂ occupies a smaller space at the H⁺ outlet than the C3-phosphate of PI(3,5)P₂. The C4-phosphate of PI(4,5)P₂ is shown as spheres.

Residues Glu55 and Ser56 are responsible for the interactions between ATP and NTD in AtCLCa (Fig. 4c), and mutation of these two residues to Ala (E55A/S56A) can completely abolish the inhibitory effect of ATP (Fig. 4h). Residue Asp753 coordinates ATP by contributing crucial hydrogen bonds (Fig. 4c), and D753A mutant impairs the inhibition of ATP (Fig. 4i), which is consistent with the previous data in plant vacuoles[41]. The structural and electrophysiological data strongly suggest that ATP stabilizes the conformation of the NTD β-hairpin to block the anion transport pathway, thus inhibiting the AtCLCa activity.

It is notable that the NTD β-hairpin is disordered in the AtCLCa-NO₃⁻ structure, and comparisons of the ATP-binding site and cytoplasmic entrance of the anion transport pathway show high similarity in AtCLCa-Cl⁻ and AtCLCa-NO₃⁻ structures and recently published AtCLCa structures[25] (Supplementary Fig. 5d–f). The previous studies showed that ATP inhibits both Cl⁻ and NO₃⁻ transport in plant vacuoles[41], and our data demonstrated that the NTD β-hairpin is the key for ATP inhibition on AtCLCa-mediated NO₃⁻ currents. All above suggested that the missing of NTD β-hairpin in AtCLCa-NO₃⁻ structure may be due to the relatively moderate resolution or different buffer environment, which occasionally happens during structure determination.

We next carried out sequence analyses of plant CLCs to check the presence of the ATP-binding site and the NTD β-hairpin, which are required for ATP binding and inhibition in AtCLCa. The result suggests that residues constituting the β-hairpin and ATP-binding site are highly conserved among the tonoplast-localized AtCLCa–c and AtCLCg, but not conserved in the Golgi- or thylakoid-localized AtCLCs (Supplementary Fig. 7a). Similar results are also found in CLC homologs that

supposed to be localized in the tonoplast of various plants, and even in *Chlamydomonas* CLC homolog (Supplementary Fig. 7b), indicating that they may share a similar regulatory mechanism by nucleotides.

## PIP₂-binding site and possible inhibition mechanism

Two significant non-protein density peaks were found on either side of the dimer interface of TMD in both AtCLCa structures. The shape of the density matched the feature of phosphatidylinositol, and the well-resolved head group allowed us to build a PI(4,5)P₂ in each protomer (Fig. 5a, Supplementary Fig. 8a). The two acyl chains of PI(4,5)P₂ are sandwiched between transmembrane helices αB and αI from one protomer and αR' from the other, which form extensive hydrophobic interactions (Fig. 5b). The inositol head group of PI(4,5)P₂ is bound in a positive-charged cavity and forms several hydrophilic interactions (Fig. 5a, b). In particular, the C1-phosphate forms hydrogen bonds with side chains of Lys86, Trp87, and Arg282 from protomer A; the C4-phosphate forms hydrogen bonds with side chains of Arg72 from protomer A and Asn579' from protomer B; the C5-phosphate forms hydrogen bonds with side chains of Trp71, Arg72, and Arg276 from protomer A. The carbonyl groups of the acyl chains of PI(4,5)P₂ form hydrogen bonds with the side chain of Trp281 from protomer A and Thr554' and Ser558' from protomer B (Fig. 5b).

It is worth noting that PI(3,5)P₂ is a featured phospholipid of plant vacuole where it inhibits the activity of AtCLCa, with a half-inhibition concentration (IC50) of 8.5 nM for the nitrate currents mediated by AtCLCa in plant vacuole, and PI(4,5)P₂ exhibits less inhibitory capacity than PI(3,5)P₂, while PI(3,4)P₂ has almost no effect[44]. Our AtCLCa structure, together with previous studies, demonstrated that the

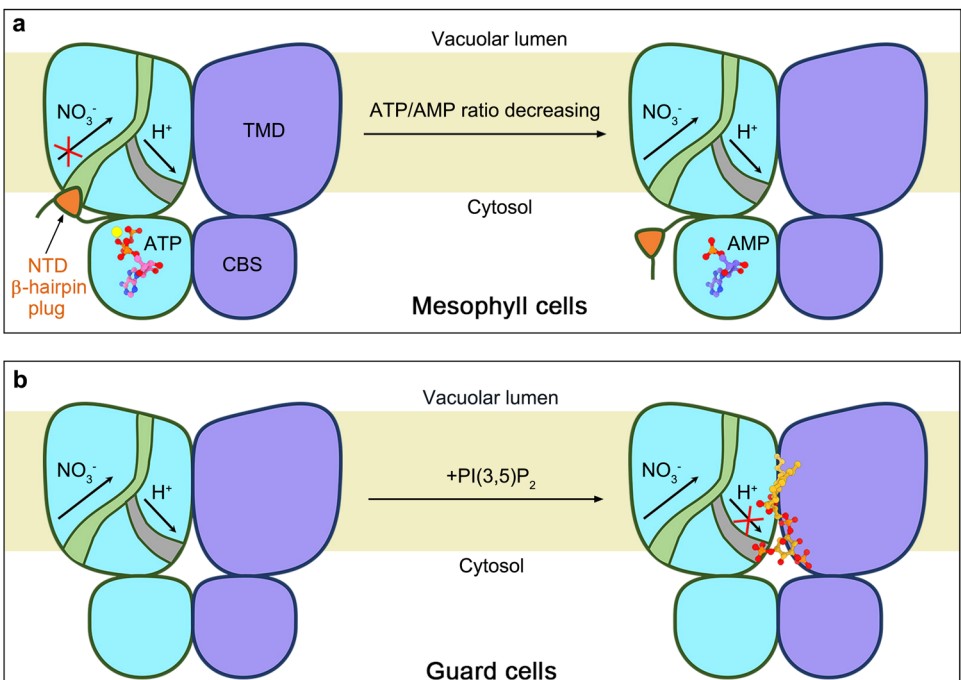

**Fig. 6 | Cartoon model for regulation of AtCLCa by nucleotides and phospholipids in different physiological scenarios. a** In mesophyll cells, AtCLCa was assumed to sense the cytosolic ATP/AMP ratio to couple the vacuolar $NO_3^-$ transport and photosynthesis. ATP functions as a molecular glue to stick the NTD with CBS domains that stabilizes the conformation of the NTD β-hairpin of AtCLCa, which may serve as a "plug" (organe-filled triangle) to block the anion transport pathway, therefore inhibiting the activity AtCLCa. AMP has little inhibition effect since it lacks the β/γ-phosphate groups which are required for the stabilization of NTD β-hairpin. **b** In guard cells, the inhibition of AtCLCa by PI(3,5)P$_2$ contributes to ABA-induced vacuolar acidification and stomatal closure. PI(3,5)P$_2$ can bind to the dimer interface of AtCLCa to occupy the H$^+$ outlet pathway, thus may inhibit the activity of AtCLCa by blocking the coupled H$^+$ transport.

specific binding of PI(4,5)P$_2$ in AtCLCa is mainly contributed by the C5 phosphate group of PI(4,5)P$_2$ (Fig. 5b), while PI(3,5)P$_2$ differs from PI(4,5)P$_2$ only in the inositol C3/C4-phosphate groups, which allows us to model a PI(3,5)P$_2$ molecule into the PI(4,5)P$_2$-binding site of AtCLCa to gain insights into its inhibitory effect (Fig. 5c). We also compared our PIP$_2$-bound AtCLCa structure with the recently reported PIP$_2$-free AtCLCa structure[25], and found that the PIP$_2$ binding did not induce conformational change of AtCLCa (Supplementary Fig. 8b, c), which excludes the allosteric inhibition regulation of AtCLCa by PI(3,5)P$_2$/PI(4,5)P$_2$.

In the modeled PI(3,5)P$_2$-bound AtCLCa structure, the inositol head group coordinated by one protomer is at the vicinity of the cytoplasmic H$^+$ outlet of the other protomer, and the free C3-phosphate of PI(3,5)P$_2$ points to the proton glutamate Glu270 (Fig. 5c, d). Since the anion transport of AtCLCa depends on the H$^+$-coupled anti-transport and the proton glutamate mutant E270A abolished both H$^+$ and anion transport[29], we suggest that the binding of PI(3,5)P$_2$ to the dimer interfaces of the TMD may occupy the cytoplasmic outlet of H$^+$ transport pathway, therefore attenuate the AtCLCa activity (Fig. 5d). The C3-phosphate of PI(3,5)P$_2$ occupies a much larger space at the H$^+$ outlet in AtCLCa structure than the C4-phosphate of PI(4,5)P$_2$ (Fig. 5d, e), which may explain the previous results that the AtCLCa-mediated vacuolar inward Cl$^-$ currents were completely abolished by 200 nM PI(3,5)P$_2$, while the same concentration of PI(4,5)P$_2$ yielded only ~60% inhibition[44]. Interestingly, most of the PIP$_2$-binding site residues are conserved in tonoplast-located AtCLCs (AtCLCa–c, g), as well as in other plant CLC homologs (Supplementary Fig. 7), suggesting that they may share a similar regulatory mechanism by phosphatidylinositol.

## Discussion

The activity of AtCLCa and many other eukaryotic CLCs, has been reported to be regulated by nucleotides and phospholipids[35–41,43,44],

pathway, therefore inhibiting the activity AtCLCa. AMP has little inhibition effect since it lacks the β/γ-phosphate groups which are required for the stabilization of NTD β-hairpin. **b** In guard cells, the inhibition of AtCLCa by PI(3,5)P$_2$ contributes to ABA-induced vacuolar acidification and stomatal closure. PI(3,5)P$_2$ can bind to the dimer interface of AtCLCa to occupy the H$^+$ outlet pathway, thus may inhibit the activity of AtCLCa by blocking the coupled H$^+$ transport.

and the regulatory mechanisms are unclear until now. On one hand, ATP inhibits the activity of AtCLCa, whose physiological function was assumed to sense the cytosolic ATP/AMP ratio to couple the vacuolar $NO_3^-$ transport and photosynthesis in mesophyll cells[41]. Our data demonstrate that ATP functions as a molecular glue to stick the NTD with CBS domains that stabilizes the conformation of the NTD β-hairpin of AtCLCa, which may serve as a "plug" to block the anion transport pathway, therefore inhibiting the activity AtCLCa. While AMP has little inhibition effect since it lacks the β/γ-phosphate groups which are required for the stabilization of NTD β-hairpin (Fig. 6a).

On the other hand, the activity of AtCLCa is also inhibited either by PI(3,5)P$_2$ or PI(4,5)P$_2$, and the former was considered to facilitate the ABA-induced guard cell vacuolar acidification and stomatal closure[5,6,42,44]. Our data together with previous studies demonstrated that the specific binding of PIP$_2$ to AtCLCa is mainly contributed by the C5 phosphate group, while PI3P may not bind to the dimer interface of AtCLCa due to lack of C5 phosphate group, even though it is the most abundant phosphatidylinositol in the vacuole membrane[46]. The structural data presented here suggest that PI(3,5)P$_2$ or PI(4,5)P$_2$ can bind to the dimer interface of AtCLCa and may inhibit the AtCLCa activity by blocking the coupled H$^+$ transport (Fig. 6b).

It is worth to note that the plant tonoplast-localized CLCs contain high sequence similarity in the residues constituting the ATP and PIP$_2$ binding sites, suggesting that these CLCs may also be regulated by these small molecules via similar mechanism. However, the regulatory effect of nucleotides differs among mammalian CLCs[36–40]. The activity of CLC-1/2 is inhibited by ATP[36–38], while that of CLC-3/4/5 is enhanced by ATP[39,40]. Of particular interest is the lysosome-localized CLC-7, whose activity is not regulated by ATP[47], though it contains a similar ATP-binding site as AtCLCa[23] (Supplementary Figs. 1, 9a, b). Therefore, the regulatory mechanism of ATP toward AtCLCa may not be applicable to all CLCs. Interestingly, most of the mammalian CLCs harbor an N-terminal domain variable in lengths and amino acids

(Supplementary Fig. 1), and it is necessary to explore the functional roles of NTDs in nucleotide regulation toward these CLCs.

In addition, the activity of CLC-7 is also attenuated by PI(3,5)P$_2$, but not PI3P[43], while the latter was identified in the CLC-7 structure[23]. Interestingly, the PI3P molecule binds to the peripheral side of CLC-7 dimer[23], which is quite different from the PIP$_2$-binding site in our AtCLCa structure (Supplementary Fig. 9a, c, d). Therefore, it is meaningful to investigate whether PI(3,5)P$_2$ also binds to the dimer interface of CLC-7 and functions by using a similar mechanism. In summary, our work provides molecular explanations for the regulation of AtCLCa by adenosine nucleotides and phospholipids, and supplies new perspectives for future studies on CLCs.

## Methods

### Protein expression and purification

The full-length gene encoding *Arabidopsis thaliana* CLCa (Uniprot: P92941) was cloned into a modified pPICZ-C vector with a C-terminal GFP-6×histidine tag fused via a short linker containing a TEV protease site. The plasmids were linearized by *Pme*I restriction enzyme and transformed into *Pichia pastoris* strain GS115 by electroporation, and positive transformants were selected on YPD plates with 1 mg/mL zeocin. For protein expression, the transformed cells were grown in MGYH medium (1.34% YNB, 1% glycerol, $4 \times 10^{-5}$% biotin, 0.004% histidine) to an $OD_{600} = 3.0-6.0$ and then induced in MMH medium (1.34% YNB, 0.5% methanol, $4 \times 10^{-5}$% biotin, 0.004% histidine) at 30 °C for an additional 48 h. Cells were harvested by centrifugation, flash-frozen in liquid nitrogen and stored at −80°C until use.

For AtCLCa-Cl$^-$, the cells were resuspended in buffer A (20 mM Tris-HCl, pH 8.0, 150 mM NaCl) and lysed using a French press. All subsequent purification steps were carried out at 4 °C. Whole-cell lysate was centrifuged at $10,000 \times g$ for 20 min and the supernatant was ultracentrifuged at $150,000 \times g$ for 1 h to pellet the membrane. The membrane fraction was resuspended in buffer A containing 1% (w/v) n-Dodecyl-β-D-Maltopyranoside (DDM, Anatrace) and 0.2% (w/v) Cholesterol Hemisuccinate Tris Salt (CHS, Anatrace) and then gently rotated at 4 °C for 2 h. After another centrifugation step at $20,000 \times g$ for 1 h, the supernatant was incubated with 5 mL Sepharose resin coupled to anti-GFP nanobody for 1 h with gentle rotation. The resin was collected in a column and washed with 100 mL Buffer A containing 0.03% DDM and 0.006% CHS, followed by TEV digestion and gently rotated overnight. Cleaved AtCLCa was eluted and concentrated (100 kDa MWKO centrifuge concentrator, Millipore) to ~1 mL and further purified by size exclusion chromatography (Superdex 200 Increase 10/300 column, GE Healthcare) in Buffer A containing 0.03% DDM and 0.006% CHS. The peak fractions were collected and concentrated for nanodisc reconstitution.

For AtCLCa-NO$_3^-$, the purified buffer A was replaced by 20 mM Tris-HNO$_3$, pH 8.0, 150 mM NaNO$_3$.

### Nanodisc reconstitution

Membrane scaffold protein MSP1D1 was expressed in *E. coli* BL21(DE3) and purified through Ni-NTA resin. Soybean polar lipid extract (Avanti) was solubilized in chloroform, dried under argon gas to form a thin lipid film. The lipid film was hydrated at a concentration of 10 mM in buffer A containing 100 mM sodium cholate. MSP1D1 and soybean polar lipid were mixed with AtCLCa using a molar ratio of 1:5:200 (AtCLCa-dimer: MSP1D1: soybean polar lipid) in buffer A with a final concentration 20 mM sodium cholate and incubated at 4 °C for 0.5 h. To remove detergent, Bio-Beads SM2 (Bio-Rad) were then added to the mixture (in a ratio of 500 mg beads per milliliter of mixture) and incubated at 4 °C for 2 h. After the second addition of Bio-Beads, the mixture was rocked gently overnight. Nanodisc-embedded AtCLCa was further purified using a Superdex 200 Increase column in buffer A. The peak fractions were collected and concentrated to 2 mg/mL for cryo-EM sample preparation.

### Cryo-EM sample preparation and data acquisition

The cryo-EM grids were prepared by applying 4 μL AtCLCa protein sample to glow-discharged holey carbon grids (Quantifoil Cu R1.2/1.3 300 mesh) and were blotted for 3 s under 100% humidity at 8 °C before plunge-frozen in liquid ethane cooled by liquid nitrogen using a Vitrobot Mark IV (Thermo Fisher Scientific). The grids were loaded into a Titan Krios (Thermo Fisher Scientific) electron microscope operating at 300 kV, equipped with the GIF Quantum energy filter and a K3 direct electron detector (Gatan). EPU (Thermo Fisher Scientific) or SerialEM[48] were used for automated data collection in the super-resolution mode.

For AtCLCa-Cl$^-$, a nominal magnification of ×105,000 was used for imaging, yielding a pixel size of 0.83 Å on images. The defocus range was set from −1.2 to −2.2 μm. Image stacks were acquired with an exposure time of 2 s and dose-fractionated to 40 frames with a total dose rate of 56.3 e$^-$ Å$^{-2}$.

For AtCLCa-NO$_3^-$, a nominal magnification of ×105,000 was used for imaging, yielding a pixel size of 0.832 Å on images. The defocus range was set from −1.2 to −2.2 μm. Image stacks were acquired with an exposure time of 2 s and dose-fractionated to 40 frames with a total dose rate of 52 e$^-$ Å$^{-2}$.

### Image processing

All movie images were aligned with MotionCorr2[49] and the CTF parameters were estimated with Gctf[50]. Other steps of image processing were performed using RELION (v.3.0)[51] or cryoSPARC (v2.15)[52].

For AtCLCa-Cl$^-$, 876,759 particles were automatically picked using RELION from 2,666 micrographs. After two rounds of two-dimensional (2D) classifications, 278,783 particles were selected to generate an initial model by RELION. Four classes were generated after three-dimensional (3D) classification, and the best representative class with 135,900 particles were selected for 3D refinement and postprocessing, resulting in a 3.11-Å map. The application of C2 symmetry and CTF refinement improved resolution to 2.96 Å. The overall resolutions were estimated based on the gold-standard Fourier shell correlation = 0.143 criterion. Local resolution estimation was calculated using RELION. The details related to data processing are summarized in Supplementary Fig. 2 and Supplementary Table 1.

For AtCLCa-NO$_3^-$, 2,178,441 particles were automatically picked using cryoSPARC from 3,666 micrographs. After 2D classification, 186,810 particles were selected to build ab-initio reconstruction. After heterogenous refinement, the remaining 92,708 particles were used for non-uniform refinement with C2 symmetry imposed and resulted in a 3.16 Å map. Local resolution estimation was calculated using cryoSPARC. The details related to data processing are summarized in Supplementary Fig. 2 and Supplementary Table 1.

### Model building and refinement

The AtCLCa atomic models were built using the predicted model of AlphaFold II[53] as a starting template. The predicted structure was docked into the density map and manually adjusted and rebuilt by COOT[54]. The resulting model was refined using real_space_refine module of PHENIX[55] with secondary structure and geometry restraints. The structures were validated with MolProbity[56], and refinement statistics can be found in Supplementary Table 1. All structure figures were prepared in PyMOL (Schrödinger) and ChimeraX[57].

### Electrophysiology

HEK293T cell line (Catalog number SCSP-502) was ordered from the cell bank of the Chinese Academy of Sciences. All the AtCLCa constructs were subcloned into the pKH3 vector and then transiently transfected into HEK293T cells using lipofectamine 3000 (Invitrogen). HEK293T cells were cultured in Dulbecco's modified Eagle's medium (Corning) with 10% FBS (Gibco), and 1% penicillin/streptomycin (Gibco) at 37 °C in a 5% CO$_2$ incubator. Whole-cell patch-clamp was performed 36−48 h after transfection at room

temperature using HEKA EPC10 amplifier. Currents were evoked by clamping the cells for 2 s to voltages between −80 and 80 mV in 20 mV steps followed by a repolarizing step to −80 mV for 500 ms. The pipette solution consisted of (in mM) 110 CsCl, 10 NaCl, 5 MgCl₂, 1 EGTA, and 15 HEPES (pH 7.4 with CsOH). The chloride bath solution consisted of (in mM) 145 NaCl, 5 KCl, 1 CaCl₂, 1 MgCl₂, 20 glucose and 15 HEPES (pH 7.4 with NaOH), and the nitrate bath solution consisted of (in mM) 145 NaNO₃, 5 KCl, 1 CaCl₂, 1 MgCl₂, 20 glucose and 15 HEPES (pH 7.4 with NaOH). Adenine nucleotides (Sigma-Aldrich) were added to the pipette solution shortly before the experiments and the pH was additionally adjusted. The amplitudes of currents and the membrane capacitances of the recorded cells were measured using PatchMaster software (HEKA).

### Statistics and reproducibility
Identical batches of HEK293T cells were randomly assigned to transfect each tested construct. The transfected HEK293T cells were selected for recording by detecting their labeled fluorescence. Non-fluorescent cells and cells that did not establish a GΩ seal or lost during recording were excluded from measurements. One current trace was recorded from each cell, and the trace recorded from different cells transfected with the same construct was considered a bioduplicate (reported as *n* values). No statistical methods were used to pre-determine sample sizes. Each tested construct had been successfully repeated with at least two batches with more than 10 cells tested and all results were similar. The number was selected based on previous studies for the sample size needed to result in statistically relevant comparisons and was sufficient for performing the statistical tests. Statistical analysis was performed using two-tailed *t*-tests in GraphPad Prism 9.0.

### Reporting summary
Further information on research design is available in the Nature Portfolio Reporting Summary linked to this article.

## Data availability
The cryo-EM maps have been deposited in the Electron Microscopy Data Bank (EMDB) under the accession codes EMD-35299 (AtCLCa-Cl⁻) and EMD-35300 (AtCLCa-NO₃⁻). The coordinates have been deposited in the Protein Data Bank (PDB) under the accession codes 8IAB (AtCLCa-Cl⁻) and 8IAD (AtCLCa-NO₃⁻). Previously solved structures mentioned in this study are under the accession codes in PDB: 1OTS (CLC-ec1), 1OTU (CLC-ec1 E148Q mutant), 3ORG (CmCLC), 6COY (HsCLC-1), 6V2J (CLC-ec1 QQQ mutant), 7JM7 (HsCLC-7/OSTM1) and 7XA9 (AtCLCa). Source data are provided with this paper.

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

## Acknowledgements

We thank Dr. Mingming Zhang at the cryo-EM center of the CAS Interdisciplinary Research Center on Biology and Chemistry, and Dr. Anqi Dong and Hui Zhao at the cryo-EM center of Fudan University for their technical assistance on cryo-EM data collection. We also thank Dr. Minhua Zhang at the core facility of Center for Excellence in Molecular Plant Sciences. This work was supported by grants from the National Natural Science Foundation of China (32025020 and 32230050 to P.Z., 91949206 to S.C., 31970146 to Z.C., and 32100961 to X.Z.) and the Chinese Academy of Sciences (CAS) (XDB27020103 and 317GJHZ2022023GC to P.Z.).

## Author contributions

Z.Y., X.Z., and S.Y. designed and performed the bulk of the experiments. Z.Y. carried out protein expression and purification, grid sample preparation, and analysis. X.Z. and Z.Y. carried out cryo-EM data collection and structure determination supervised by Z.C. and P.Z. S.Y. and Z.Y. carried out electrophysiological experiments guided by S.C. J.Z., X.H., and F.Y. contributed to protein purification and grid sample preparation. P.Z. and Z.Y. wrote the manuscript with inputs from other authors. P.Z. conceived the project.

## Competing interests

The authors declare no competing interests.
