## [Peer Review File · Nature Communications]

Molecular mechanism underlying regulation of Arabidopsis CLCa transporter by nucleotides and phospholipidsEditorial Note: This manuscript has been previously reviewed at another journal that is not operating a transparent peer review scheme. This document only contains reviewer comments and rebuttal letters for versions considered at *Nature Communications*.

REVIEWERS' COMMENTS

Reviewer #1 (Remarks to the Author):

In their study, Zhao Yang et al. revealed cryo-EM structures of the plant nitrate/proton exchanger CLCa in complex with different regulatory molecules. They identified the transport pathways for chloride, nitrate and protons. Novel structural features allowed to propose possible mechanisms for the previously described inhibitory action of ATP and phosphoinositides. Structural data were supported by site-directed mutagenesis and patch-clamp electrophysiology. The study is clearly written and presented and generally convincing. I have only three points of criticism.

First, site-directed mutagenesis is not exhaustive, especially for the part on phosphoinositides. More structural and functional data could make the conclusions stronger and more convincing.

Second, the authors should comment on the accessibility of bulk membrane lipids to the phosphoinositide binding site at the dimer interface. PI(3,5)P₂ is a very-low-abundance phosphoinositide which is produced under very limited conditions by specific kinases, while its precursor PI3P is quite abundant in the vacuolar membrane. Can bound PI3P possibly be directly phosphorylated at the binding site or is a dynamic exchange of phosphoinositide species more likely?

Third, the summary sketches in Figure 6 are poor and should be removed from the manuscript.

Reviewer #3 (Remarks to the Author):

As the fifth reviewer of this paper, I would not only be giving my opinion on the paper, but also will be judging it holistically in context with their response to the other reviewer's comments. I will avoid reiterating points brought up by the existing four reviewers.

Yang et al. presents the first substrate bound structures of Arabidopsis CLCa transporter bound to Cl⁻ and NO₃⁻. Together with electrophysiological data, they suggest the regulatory mechanism of AtCLCa by nucleotides and phospholipids under certain physiological scenarios. While structural and functional work was done admirably, many of the conclusions overlap with the Jin et al. JBC paper as all four other reviewers pointed out. The novel conclusions pertain to the presence of the ions, regulatory mechanism of PIP₂ and additional resolved N-terminal beta-hairpin.

In terms of response to the four reviewers, the authors have made an excellent effort. In my opinion, they have revamped the manuscript to eliminate as far as possible any ambiguous or erroneous statements and cited more of the relevant literature that was missing in the initial manuscript.

Hence my recommendation is for this paper to be passed to Communications Biology for

immediate publication.

Minor

The "2" in PIP2 should be subscript, i.e. PIP₂

I thank the authors for taking the time to answer all my questions about their analyses on CLCa structure, a nitrate/proton exchanger involved in nitrate storage inside the vacuole in Arabidopsis cells. In this new version of the manuscript, the authors have put their work in the context by citing the correct literature and removing some results. However, even if this new version is much clearer, I remain sceptical about the novelty brought by these results to be published in Nature Communications for the following reasons:

1. He et al (JBC, 2023) have already published the function of N-terminal domain of CLCa in the regulation by ATP. The authors of the submitted manuscript have just confirmed the importance of E55 and S56 in this regulation by characterizing mutated forms of CLCa by electrophysiology.
2. The regulation by phospholipid is new, but still remains very descriptive. The authors argue that comparing their results to the PIP2-free structure obtained by He et al (2023), PIP2 do not induce conformational change and just appears to block the H⁺ proton pathway of CLCa. However, it would have been more informative to analyse if the interaction with some amino acids and PIP2 are important for this blockage by directed mutagenesis.
3. The authors argue that the differences obtained in electrophysiology on native vacuole (De Angeli et al Nature 2006) and with their HEK cells-using system might be due to differences of the used solutions (Extended data Fig 6e). In this experiment, the shift in Erev is much smaller than the ones previously described. It will have been good to include statistical analyses and the numbers of cells analysed, so that the readers/reviewers could check the solidity of the results. This should be extended to all the electrophysiological analyses.

REVIEWERS' COMMENTS

Reviewer #1 (Remarks to the Author):

In their study, Zhao Yang et al. revealed cryo-EM structures of the plant nitrate/proton exchanger CLCa in complex with different regulatory molecules. They identified the transport pathways for chloride, nitrate and protons. Novel structural features allowed to propose possible mechanisms for the previously described inhibitory action of ATP and phosphoinositides. Structural data were supported by site-directed mutagenesis and patch-clamp electrophysiology. The study is clearly written and presented and generally convincing. I have only three points of criticism.

First, site-directed mutagenesis is not exhaustive, especially for the part on phosphoinositides. More structural and functional data could make the conclusions stronger and more convincing.

RE: We tried to carry out inhibitory analysis of PIP₂ toward AtCLCa in HEK293T cells using both whole-cell patch clamp and inside-out patches before first submission and during revision, but not succeeded. The probable reason may be that the intracellular membrane (plant vacuole) located AtCLCa by nature has small current amplitudes which may preclude inside-out-patch measurement when heterologously expressed in the plasma membrane of HEK293T cells. Similar results have been observed in CLC-5 analysis (Grieschat et al, EMBO reports, 2020).

We agree with this reviewer that the PIP₂ inhibition on AtCLCa needs further functional verification, therefore, we revised the manuscript by soften the tones.

Second, the authors should comment on the accessibility of bulk membrane lipids to the phosphoinositide binding site at the dimer interface. PI(3,5)P₂ is a very-low-abundance phosphoinositide which is produced under very limited conditions by specific kinases, while its precursor PI3P is quite abundant in the vacuolar membrane. Can bound PI3P possibly be directly phosphorylated at the binding site or is a dynamic exchange of phosphoinositide species more likely?

RE: We thank this reviewer for this question.

PI(3,5)P₂ is a kind of phosphatidylinositol with very low abundance. Although the presence of PI(3,5)P₂ was not detected in the vacuolar membrane by fluorescent marker (Simon et al, the plant journal 2014), it was found that a class of phosphatase was located on the vacuolar membrane and was responsible for dephosphorylation of PI(3,5)P₂ to generate PI3P (Nováková et al, PNAS 2014), thus confirming the location of PI(3,5)P₂ in the vacuolar membrane.

The synthesis of PI(3,5)P₂ can be induced by hyperosmotic stress, such as drought (Meijer et al, Planta, 1999). Under drought stress, plants produce ABA and induce stomatal closure to save water, a process that is accompanied by vacuole acidification and convolution of guard cells which requires PI(3,5)P₂ (Bak et al, The plant cell

2013).

PI(3,5)P₂ can inhibit the activity of AtCLCa in the vacuole of mesophyll cells. The inhibitory effects of different PIP₂ toward plant vacuolar AtCLCa have been thoroughly studied (Carpaneto et al, EMBO Rep 2017), in which PI(3,5)P₂ could inhibit AtCLCa with an IC₅₀ of 11.7 nM, and PI(4,5)P₂ exhibit less inhibitory capacity than PI(3,5)P₂, while PI(3,4)P₂ has almost no inhibitory effect. Therefore, it is reasonable to assume that the C5 phosphate group of PIP₂ is required for AtCLCa inhibition.

Our AtCLCa structure, together with previous studies, demonstrated that the specific binding of PI(4,5)P₂ in AtCLCa is mainly contributed by the C5 phosphate group of PI(4,5)P₂. To our opinion, PI3P may not bind to the dimer interface of AtCLCa due to lack of C5 phosphate group, even though it is the most abundant phosphatidylinositol in the vacuole membrane. On the other hand, although both PI(4,5)P₂ and PI(3,5)P₂ can inhibit AtCLCa activity, PI(4,5)P₂ is only localized to the plasma membrane, therefore, under physiological conditions, only PI(3,5)P₂ among the phosphatidylinositol can bind and inhibit AtCLCa activity in the vacuolar membrane.

We accepted this reviewer's advice by adding further discussion to the revised manuscript in line 338-342: "Our data together with previous studies demonstrated that the specific binding of PIP₂ to AtCLCa is mainly contributed by the C5 phosphate group, while PI3P may not bind to the dimer interface of AtCLCa due to lack of C5 phosphate group, even though it is the most abundant phosphatidylinositol in the vacuole membrane."

Third, the summary sketches in Figure 6 are poor and should be removed from the manuscript.

RE: We think that this working model is very important for the understanding of the whole manuscript. This model incorporates the mechanistic finding of this work--regulatory mechanism of AtCLCa activity by nucleotides and phospholipids--into the corresponding physiological scenarios, which highlights the physiological significance of our work.

Reviewer #2 (Remarks to the Author):

I thank the authors for taking the time to answer all my questions about their analyses on CLCa structure, a nitrate/proton exchanger involved in nitrate storage inside the vacuole in Arabidopsis cells. In this new version of the manuscript, the authors have put their work in the context by citing the correct literature and removing some results. However, even if this new version is much clearer, I remain sceptical about the novelty brought by these results to be published in Nature Communications for the following reasons:

1. He et al (JBC, 2023) have already published the function of N-terminal domain of CLCa in the regulation by ATP. The authors of the submitted manuscript have just confirmed the importance of E55 and S56 in this regulation by characterizing mutated forms of CLCa by electrophysiology.

Compared with our AtCLCa-Cl⁻ structure, the AtCLCa structure published by He et al (JBC, 2023) lacks the N-terminal β -hairpin (residues 44-53). More importantly, the N-terminal β -hairpin is stabilized by ATP binding to block the anion transport pathway, thereby inhibiting the AtCLCa activity according to our structural and electrophysiological analyses. Residues E55 and S56 are responsible for the interactions between ATP and N-terminal domain in AtCLCa, and E55A/S56A double mutant abolished the ATP inhibition.

Even if E55 and S56 of AtCLCa were found to be involved in ATP binding, no conclusion about the regulatory mechanism of AtCLCa by ATP can be drawn based on the JBC-structure or literatures due to the lack of the key N-terminal β hairpin.

Therefore, the main finding of our work not only confirms the importance of E55 and S56, but more importantly, reveals that AtCLCa contains an N-terminal β hairpin which is essential for the inhibition by ATP.

2. The regulation by phospholipid is new, but still remains very descriptive. The authors argue that comparing their results to the PIP2-free structure obtained by He et al (2023), PIP2 do not induce conformational change and just appears to block the H⁺ proton pathway of CLCa. However, it would have been more informative to analyse if the interaction with some amino acids and PIP2 are important for this blockage by directed mutagenesis.

RE: We tried to carry out inhibitory analysis of PIP2 toward AtCLCa in HEK293T cells using both whole-cell patch clamp and inside-out patches before first submission and during revision, but not succeeded. The probable reason may be that the intracellular membrane (plant vacuole) located AtCLCa by nature has small current amplitudes which may preclude inside-out-patch measurement when heterologously expressed in the plasma membrane of HEK293T cells. Similar results have been

observed in CLC-5 analysis (Grieschat et al, EMBO reports, 2020).

We agree with this reviewer that the PIP2 inhibition on AtCLCa needs further functional verification, therefore, we revised the manuscript by softening the tones.

3. The authors argue that the differences obtained in electrophysiology on native vacuole (De Angeli et al Nature 2006) and with their HEK cells-using system might be due to differences of the used solutions (Extended data Fig 6e). In this experiment, the shift in Erev is much smaller than the ones previously described. It will have been good to include statistical analyses and the numbers of cells analysed, so that the readers/reviewers could check the solidity of the results. This should be extended to all the electrophysiological analyses.

RE: We accepted this advice and provided the numbers of cells analysed as precise value of 'n' in the legends of figures 4d-4i and supplementary figures 6d-6f in the revised manuscript.

Reviewer #3 (Remarks to the Author):

As the fifth reviewer of this paper, I would not only be giving my opinion on the paper, but also will be judging it holistically in context with their response to the other reviewer's comments. I will avoid reiterating points brought up by the existing four reviewers.

Yang et al. presents the first substrate bound structures of Arabidopsis CLCa transporter bound to Cl⁻ and NO₃⁻. Together with electrophysiological data, they suggest the regulatory mechanism of AtCLCa by nucleotides and phospholipids under certain physiological scenarios. While structural and functional work was done admirably, many of the conclusions overlap with the Jin et al. JBC paper as all four other reviewers pointed out. The novel conclusions pertain to the presence of the ions, regulatory mechanism of PIP₂ and additional resolved N-terminal beta-hairpin.

In terms of response to the four reviewers, the authors have made an excellent effort. In my opinion, they have revamped the manuscript to eliminate as far as possible any ambiguous or erroneous statements and cited more of the relevant literature that was missing in the initial manuscript.

RE: We thank this reviewer for his or her favorable comments toward our work.

Hence my recommendation is for this paper to be passed to Communications Biology for immediate publication.

Minor

The "2" in PIP₂ should be subscript, i.e. PIP₂

RE: Yes